# Intraoperative Dexmedetomidine Decreases Postoperative Pain after Gastric Endoscopic Submucosal Dissection: A Prospective Randomized Controlled Trial

**DOI:** 10.3390/jcm12051816

**Published:** 2023-02-24

**Authors:** Xin Luo, Peishan Chen, Xinlu Chang, Yang Li, Lei Wan, Fushan Xue, Lixin An

**Affiliations:** Department of Anesthesiology, Beijing Friendship Hospital, Capital Medical University, Beijing 100050, China

**Keywords:** endoscopic submucosal dissection, dexmedetomidine, postoperative pain, adverse events

## Abstract

Background: Postoperative pain is one of the most common complications after gastric endoscopic submucosal dissection (ESD); however, there have been only a few studies assessing the efficacy of interventions on postoperative pain after gastric ESD. This prospective randomized controlled trial was designed to assess the effect of intraoperative dexmedetomidine (DEX) on postoperative pain after gastric ESD. Materials and methods: A total of 60 patients undergoing elective gastric ESD under general anesthesia were randomly divided into the DEX group receiving DEX with a loading dose of 1 μg/kg, followed by a maintenance dose of 0.6 μg/kg/h until 30 min before the end of the endoscopic procedure, and the control group receiving normal saline. The primary outcome was the visual analog scale (VAS) score of postoperative pain. Secondary outcomes were the dosage of morphine for postoperative pain control, hemodynamic changes during the observable period, the occurrence of adverse events, lengths of postanesthesia care unit (PACU) and hospital stays, and patient satisfaction. Results: The incidence of postoperative moderate to severe pain was 27% and 53% in the DEX and control groups, respectively, with a significant difference. Compared to the control group, VAS pain scores at 1 h, 2 h, and 4 h postoperatively, the dosage of morphine in the PACU, and the total dosage of morphine within 24 h postoperatively were significantly decreased in the DEX group. Both incidences of hypotension and use of ephedrine in the DEX group were significantly decreased during surgery, but they were significantly increased in the postoperative period. Postoperative nausea and vomiting scores were decreased in the DEX group; however, the length of PACU stay, patient satisfaction, and duration of hospital stay were not significantly different between groups. Conclusion: Intraoperative DEX can significantly decrease postoperative pain level, with a slightly reduced dosage of morphine and a decreased severity of postoperative nausea and vomiting after gastric ESD.

## 1. Introduction

Endoscopic submucosal dissection (ESD) has become a standard treatment of early gastrointestinal neoplasms as it can completely remove lesions, allow accurate evaluation of the histopathological curability, and reduce postoperative loss of function [1]. However, it is reported that ESD has a high incidence of complications, such as bleeding, perforation, stenosis, fever, abdominal distension, postoperative pain, and others [2,3]. Of these complications, postoperative pain is the most easily underestimated and ignored by endoscopists and anesthesiologists. Postoperative pain can not only result in a direct negative impact on patient satisfaction but also may prolong hospital stays and increase hospital expenses [2,3,4]. Furthermore, patients would often need a second ESD procedure even with complete lesion resection in the first treatment [5]. Additionally, patients with a painful or unhappy experience at the first treatment are often afraid of a second endoscopic procedure. All of those indicate that effective postoperative pain control after an ESD procedure is important for patients’ short-term outcomes and long-term treatment compliance. In clinical practice, however, both anesthesiologists and clinicians are often reluctant to use painkillers such as opioid drugs due to the concern that these drugs may mask some postoperative complications of ESD, such as perforation, hemorrhage, and others [6,7]. In the available literature, thus, there have only been a few studies regarding postoperative pain management after ESD [8,9,10]; further studies are needed to obtain the appropriate and effective methods of pain control after ESD.

Dexmedetomidine (DEX) is a novel selective α2 agonist with sedative, anti-anxiety, and analgesic effects but without the risk of respiratory depression [11]. The available evidence indicates that the addition of DEX to a postoperative analgesia scheme can provide improved pain control with reduced risk of postoperative nausea and vomiting [12,13,14]. Nonetheless, it is still unclear whether intraoperative administration of DEX during an ESD procedure with total intravenous anesthesia can attenuate postoperative pain levels. Thus, this prospective randomized controlled trial was designed to verify the hypothesis that intraoperative DEX can reduce postoperative pain after gastric ESD and improve patients’ outcomes.

## 2. Materials and Methods

### 2.1. Study Design

This was a single-center, prospective, double-blinded, randomized study. The study protocols were approved by the Ethics Committee of Beijing Friendship Hospital, Capital Medical University (Approval No: 2021-P2-003-01), registered on the Chinese Clinical Trial Registry (https://www.chictr.org.cn/, registration number: ChiCTR2100043837), and published on Trials [15]. The recommendations of the CONSORT 2010 Statement were followed.

### 2.2. Patients

From 20 March 2021 to 31 March 2022, patients who underwent ESD for gastric neoplasms including early-stage cancer or tumor in our endoscopy center were enrolled. The inclusion criteria were 18–65 years and American Society of Anesthesiologists (ASA) physical status grade I-II. The exclusion criteria included: (1) sinus bradycardia; (2) sick sinus syndrome; (3) predicted difficult airway or obesity (body mass index > 35 kg/cm^2^); (4) mental illness; (5) allergic to DEX; (6) long-term history of opioid use. If patients requested to withdraw from the trial, refused to use analgesic drugs after surgery, or violated the protocol, they were also excluded. If the patient was transferred to open surgery, had an operation time >4 h, and needed reexamination or reoperation with endoscopy due to postoperative complications within 48 h after surgery, their data were not included in the final analysis. Written informed consent was obtained from each included patient.

### 2.3. Randomization and Masking

The included patients were randomly allocated into two groups according to the random numbers generated by the computer according to 1:1 allocation ratio: DEX and control groups. In the DEX group, DEX was diluted with 50 mL saline at a concentration of 4 µg/mL. In the control group, normal saline of the same volume was administered. The medications used for both groups were prepared by a specialized nurse using the same syringes in appearance. The results of computer-based randomization were placed in a sealed envelope and opened only when medication preparation was completed. Once a patient was included, the nurse distributed the corresponding drug to the anesthesiologist. The researchers, anesthesiologists, endoscopists, and physicians in charge of follow-up were all blinded to the grouping assignment.

### 2.4. Anesthesia

The patients underwent gastrointestinal preparation and fasted for 8 h before the endoscopic procedure according to our routine practice. After entering the endoscopic room, non-invasive blood pressure, heart rate (HR), pulse oxygen saturation (SpO_2_), and bispectral index (BIS) were routinely monitored. After intravenous access was established, the patient was placed at a lateral position where they were comfortable in awake status and preoxygenation was performed. The patients in the DEX group received an intravenous bolus of DEX 1 µg/kg in 10 min via an infusion pump, and control patients received normal saline of the same volume in 10 min. Then, remifentanil 1–2 µg/kg, propofol 1.5–2 mg/kg, and rocuronium 0.6–0.8 mg/kg were intravenously administered for anesthesia induction. After patients lost consciousness, tracheal intubation was performed, with a size 7.5 tracheal tube for males and size 6.5 for females. All intubation was performed by an attending anesthesiologist with extensive experience in airway management in a lateral position. After intubation, the lungs were mechanically ventilated. The tidal volume was set at 8 mL/kg and the respiratory rate was adjusted to maintain an end-tidal carbon oxide (P_ET_CO_2_) of 35–45 mm Hg throughout the procedure. In addition to maintenance of anesthesia by intravenous infusion of propofol and remifentanil, patients in the DEX group simultaneously received an intravenous infusion of DEX at a rate of 0.6 μg/kg/h, and control patients received normal saline at the same rate. Throughout the procedure, infusion rates of propofol and remifentanil were adjusted to keep the changes in blood pressure and HR within 20% of their baseline values, and rocuronium was administered as needed to maintain adequate muscle relaxation.

In this study, all gastric ESD procedures were performed by endoscopists with over 5-year experience in ESD and more than 500 ESD surgeries. The standard procedures included performing a submucosal injection, incision of the lateral margin of the tumor, dissection of the tumor, and electrocoagulation hemostasis. The marks were firstly made 5 mm surrounding the lesion mucosa using a needle knife. Then, the mixture of epinephrine (1:100,000 solution in saline) and indigo carmine was injected into the submucosal layer around the marks. After the lesion mucosa was removed with an electrosurgical knife from the marks, hemostasis was performed with hemostatic forceps or a clutch cutter. For each patient, carbon dioxide insufflation was used during gastric ESD and stopped immediately at the end of the procedure.

Intravenous infusion of DEX and saline was stopped 30 min before the end of the endoscopic procedure. At the end of the procedure, intravenous propofol and remifentanil were stopped, and tramadol 50 mg and ondansetron 4 mg were intravenously given as analgesic and anti-emetic drugs, respectively. As needed, atropine 0.5 mg and neostigmine 1 mg were intravenously administered to antagonize residual neuromuscular block. After spontaneous breathing was adequately resumed and the patient was able to follow the instructions, the trachea was extubated and the patient was sent to the postanesthesia care unit (PACU) for observation. The patients were discharged from the PACU after discharge criteria were achieved. A single dose of omeprazole (40 mg) was intravenously administered 2 h after surgery.

A physician who was blinded to the grouping assignment assessed postoperative pain levels by a 0–10 point visual analog scale (VAS) at 1, 2, 4, 6, 12, 24, and 48 h after surgery. The VAS score “0” was defined as no pain and “10” was defined as pain beyond imagination. According to the VAS scores, the severity of postoperative pain was classed into mild (0–3), moderate (4–6), and severe (7–10). If the VAS pain score was ≥4 or the patient needed additional analgesia, 1 mg of intravenous morphine was administered.

The age, gender, height, weight, chronic conditions, smoking status, drinking status, duration of anesthesia, operative time, bleeding volume, adverse hemodynamic events, the dosage of anesthetic and analgesic drugs, time to awakening, extubation time, and clinicopathological characteristics (including location, depth, and pathological classifications) of the lesions removed by ESD were recorded. The duration of anesthesia is the time from the beginning of anesthesia induction to the completion of extubation. The duration of the procedure is the time from the insertion of the endoscope to the completion of lesion site hemostasis. The time to awakening was the time from the termination of anesthetics to awakening. The time to extubation was the time from the termination of anesthetics to extubation. After surgery, patient satisfaction, duration of PACU stay, and length of hospital stay were also noted.

### 2.5. Outcome Endpoints

The primary endpoint of this study was VAS pain score at 2 h postoperatively. The secondary outcomes were VAS pain score at other time points (1, 4, 6, 12, 24, and 48 h postoperatively), the incidence of moderate to severe pain, nausea and vomiting scores, and incidence of nausea and vomiting score ≥2. The use of postoperative analgesic and anti-emetic drugs was noted. Furthermore, intraoperative and postoperative adverse events, including hypoxemia or apnea, and adverse hemodynamic events were also recorded. Hypoxemia and apnea are defined as SpO_2_ less than 92% and breathing apnea for >60 s, respectively [16]. Hemodynamic variables were recorded before induction (T0), at 1 min after induction (T1), at intubation (T2), at 5 min after intubation (T3), at the end of the procedure (T4), at extubation (T5) and 5 min after extubation (T6). Adverse hemodynamic events included hypertension, hypotension, tachycardia, and bradycardia. Hypertension is defined as a mean artery pressure (MAP) increase of >20% from baseline. Hypotension is defined as a MAP decrease of >20% from baseline. Tachycardia is defined as a HR of >100 beats/min, which was treated with intravenous esmolol 10 mg, if necessary. Bradycardia is defined as a HR of <45 beats/min, which was treated with intravenous atropine 0.5 mg, if necessary. If hypotension persisted for more than 2 min and did not respond to a load therapy of 200 mL fluid, 6 mg of ephedrine was administrated intravenously [17].

Nausea and vomiting severity was graded using a 4-point scale (0, no nausea and vomiting; 1, mild nausea; 2, moderate nausea; and 3, vomiting) [18]. If the nausea and vomiting score was ≥2, ondansetron 4 mg was intravenously administrated. If postoperative nausea and vomiting were not relieved by the above treatments, rescue medications could be administered again.

### 2.6. Sample Size Calculation

The sample size of this study was calculated according to the results of our pre-experiment, in which postoperative pain VAS scores at 2 h after surgery were 0.67 ± 1.63 in the DEX group and 2.38 ± 1.06 in the control group, respectively. Based on the between-group differences in the means of postoperative pain VAS scores at 2 h after surgery, with a probability of a type 1 error of 0.05 (α = 0.05), a probability of a type 2 error of 0.1 (β = 0.1), and a power of 0.90, 27 patients in each group would be needed. Considering a ratio of 1:1 grouping assignment and a 10% dropout rate, a sample size of 30 cases in each group was determined.

### 2.7. Statistical Analysis

Statistical analysis of data was performed by a statistical team of our hospital blinded to the grouping assignment with SPSS 20.0 (version 20.0; SPSS Inc., Chicago, IL, USA). For continuous variables, the Kolmogorov-Smirnov test was first applied to determine data distribution. For data with a normal distribution, they were present as means ± standard deviations (SD), and intergroup comparisons were carried out by an independent Student’s *t*-test. For data with a non-normal distribution, they were present as medians (interquartile range, IQR) and intergroup comparisons were performed by the Mann-Whitney U-test and Wilcoxon signed-rank test. The categorical variables were present as numbers and/or percentages, and intergroup comparisons were performed by a Fisher exact test or Chi-square test according to their frequency. All *p* values were unilateral, and *p* < 0.05 was considered statistically significant.

## 3. Results

### 3.1. Enrollment Process, Demographic Data, and Clinicopathological Characteristics of Included Patients

The flow chart of included and excluded patients was shown in Figure 1. From March 2021 through March 2022, a total of 194 patients were screened. Of these, 126 patients were excluded because they were not eligible for inclusion criteria (*n* = 83), refused to participate (*n* = 13), and used postoperative analgesic drugs (*n* = 30). Thus, 68 patients were equally randomized into two groups. After randomization, 1 patient in the DEX group and 2 in the control group were further excluded because they were converted to open surgery after endoscopy. A total number of 3 patients in the DEX group and 2 in the control group required reoperation due to postoperative bleeding within 48 h following the ESD surgery and their data were not taken into data analysis. Finally, 30 patients in each group were included in the data analysis. The demographic data and clinicopathological characteristics of patients are shown in Table 1 and were not significantly different between groups (*p* > 0.05).

### 3.2. Postoperative Pain Levels and Morphine Consumption

The postoperative pain levels are shown in Figure 2. The VAS pain scores at 1 h, 2 h, and 4 h postoperatively were significantly lower in the DEX group than in the control group (*p* < 0.05). However, there were no significant differences between groups in VAS pain scores at 6 h, 12 h, 24 h, and 48 h postoperatively (*p* > 0.05). The incidence of moderate to severe pain was 26.0% and 53.3% in the DEX and control groups, respectively, with a significant between-group difference (*p* < 0.05). The dosage of morphine in the PACU was significantly smaller in the DEX group than in the control group (*p* = 0.000). The dosage of morphine for postoperative pain control in the ward was not significantly different between groups (*p* > 0.05). The total consumption of morphine within 24 h after surgery was significantly increased in the control group compared with the DEX group (*p* = 0.009, Table 2).

### 3.3. Anesthesia and Intraoperative Data

The duration of anesthesia and procedure, dosages of propofol and remifentanil, volumes of intraoperative bleeding and fluid, incidences of hypertension, tachycardia, and bradycardia, and times to awakening and extubation were not significantly different between groups (*p* > 0.05). However, the incidence of hypotension and the use of ephedrine during surgery were increased in the control group compared to the DEX group (Table 3). The MAP and HR changes during the observable period are shown in Figure 3. MAPs at 1 min after induction, intubation, 5 min after intubation, and at the end of the procedure were significantly higher in the DEX group than in the control group (*p <* 0.05), but MAP at extubation was significantly increased in the control group compared to the DEX group (*p <* 0.05). HRs at intubation, extubation, and 5 min after extubation were significantly decreased in the DEX group compared to the control group (*p <* 0.05). 

### 3.4. Postoperative Adverse Events and Outcomes

In the PACU, incidences of nausea and vomiting score ≥ 2, hypertension, and bradycardia were not significantly different between groups; however, the incidence of hypotension was significantly higher in the DEX group than in the control group (*p* = 0.024). Meanwhile, nausea and vomiting scores at PACU arrival, 15 min after PACU arrival, and at PACU discharge were lower in the DEX group compared to the control group (*p* < 0.05). 

Compared with the DEX group, the postoperative dosage of ondansetron was significantly increased, and the postoperative dosage of ephedrine was significantly decreased in the control group (*p* < 0.05). However, patient satisfaction, and lengths of PACU stay and hospital stay did not significantly differ between groups (*p* > 0.05, Table 4).

## 4. Discussion

Our study was the first to assess the influence of intraoperative DEX on postoperative pain after ESD for gastric tumors. The results showed that intraoperative DEX significantly decreased postoperative pain levels and the use of analgesics for postoperative pain control. These results are in agreement with the findings of a meta-analysis for adult surgical patients with general anesthesia, in which DEX produces effective analgesia within 6 h postoperatively and decreases total opioid consumption within 24 h postoperatively [19].

The available literature indicates that a loading dose of DEX 1 µg/kg followed by intravenous infusion of DEX at a rate of 0.2–0.7 µg/kg/h was effective and safe for both sedation in ICU patients and anesthesia in surgical adults [20]. Therefore, a loading dose of 1 µg/kg and a maintenance infusion rate of 0.6 µg/kg/h were selected in our study. Our results showed that pain scores at 1 h, 2 h, and 4 h postoperatively were significantly decreased in the DEX group. Furthermore, the dosage of morphine during the PACU stay and the dosage of morphine within 24 h postoperatively were also significantly decreased in the DEX group compared to the control group. All of these indicate that intraoperative DEX is beneficial to early postoperative pain control after gastric ESD.

DEX is a highly selective α2-receptor agonist with characteristics of sedation, analgesia, anti-anxiety, antiemesis, and sympathetic inhibition [21,22,23]. The available literature indicates that DEX activates α2 receptors in the brain and anterior horn of the spinal cord and thereby acts as a central analgesic. In the peripheral nervous system, moreover, DEX can inhibit the activation of nociceptive neurons related to A δ and C fibers and thereby act as a non-opioid peripheral analgesic [24]. Recent studies have shown that DEX has a special effect on relieving visceral pain [25,26,27,28]. In patients undergoing laparoscopic gastrointestinal surgery, Jang et al. [28] confirmed that compared with hydrocodone alone, DEX effectively reduced postoperative visceral pain and improved the quality of sleep. Postoperative pain was one of the most common complications after gastric ESD, with an incidence of up to 98% [9] and the incidence of moderate to severe pain was as high as 36.4–42.3% [4,10]. Recent studies show that pain after gastric ESD is attributable to transmural burn, air leakage, high acid sensitivity, and inflammatory response. Effective treatments for postoperative pain include dexamethasone, percutaneous fentanyl patch, local lidocaine injection, and proton pump inhibitors [9,29,30,31,32,33]. Lee et al. [32] found that 7% of ESD patients with postoperative pain also suffered from fever without an identifiable cause, indicating that postoperative pain after ESD may be related to inflammation. Furthermore, it has been reported that a single intravenous administration of dexamethasone can reduce the severity of postoperative pain after ESD through an anti-inflammatory effect [29]. In addition, pain after ESD is visceral pain, which is transmitted to the central nervous system through the intestinal nerves [30,32]. Thus, DEX may produce an analgesic effect after gastric ESD by complicated interaction between anti-inflammatory effect and non-opioid analgesic effect.

Postoperative nausea and vomiting may not only result in patient discomfort and dissatisfaction but also prolong the duration of the hospital stay [34]. It has been shown that a loading dose of DEX 0.6–1 µg/kg can effectively reduce the incidence of postoperative nausea and vomiting in patients undergoing surgery [35,36]. Furthermore, the antiemetic effect of DEX helps to improve the quality of postoperative recovery [18,37]. In our study, intraoperative DEX decreased the incidence and severity of nausea and vomiting scores in the PACU and reduced the dosage of antiemetics after gastric ESD. This is in accordance with the results of previous studies in surgical patients [35,36].

It is reported that a loading dose of DEX 0.6–1 µg/kg before surgery with or without a continuous intravenous infusion of maintenance dose can provide stable intraoperative hemodynamic variables in intubated patients with general anesthesia [36,38]. In our study, MAPs during anesthesia induction and intubation were higher in the DEX group than in the control group. This may be attributable to intravenous infusion of DEX loading dose. Previous work showed that rapid intravenous infusion of DEX could result in transient blood pressure elevation and HR reduction, which were alleviated after cessation of intravenous DEX infusion [39]. The intravenous infusion of DEX loading dose in the current study followed the requirements of guidelines (>10 min) [20], but blood pressure elevation still occurred in three cases during anesthesia induction. Nonetheless, blood pressure elevation was transient and automatically relieved after anesthesia induction and continuous intravenous infusion of the DEX maintenance dose. Most importantly, both the incidence of hypotension and dosage of ephedrine during the endoscopic procedure were lower in the DEX group than in the control group, suggesting that DEX can stabilize intraoperative hemodynamic variables.

It has been demonstrated that the most common adverse events associated with intravenous DEX were hypotension and bradycardia [40]. These findings were in agreement with the results of our study, in which the percentage of patients with hypotension during the PACU stay was higher in the DEX group compared with the control group (20% vs. 0%). In our study, however, there were no life-threatening adverse events associated with intravenous DEX. Furthermore, both hypotension and bradycardia were effectively treated with intravenous ephedrine and atropine.

The available evidence indicates that the best time to stop the intravenous infusion of DEX is 30 min before the end of the surgery, as it does not affect the postoperative recovery of patients and helps to decrease the occurrence of agitation in the early recovery period [41]. In our study, thus, intravenous DEX was stopped 30 min before the end of the endoscopic procedure. Our results showed that recovery time, duration of PACU stay, length of hospital stay, and patient satisfaction were not significantly different between groups, meaning that intraoperative DEX provides stable hemodynamic variables and reduces postoperative pain, but does not improve patient satisfaction or affect postoperative recovery. By further analysis of the results, we noted that DEX reduced the incidence of moderate to severe pain after gastric ESD, but the highest postoperative pain score in the two groups was 5, and the dosage of morphine for postoperative pain control in the ward was not significantly different between groups. These results suggest that some of the patients in this study experienced moderate to severe postoperative pain, but their pain was promptly treated. Indeed, the total dosage of morphine within 24 h postoperatively was significantly higher in the control group compared to the DEX group. However, the total dosage of morphine within 24 h postoperatively in the control group was very small (1 mg; IQR, 0–2.25) and no adverse events associated with morphine administration occurred. Perhaps, these are possible reasons why intraoperative DEX does not influence patient satisfaction and length of hospital stay, though it improves postoperative pain control and decreases total consumption of morphine for postoperative pain control.

## 5. Limitations

There are some limitations in the design of this study that deserve special attention. First, only one dose regimen was designed; it was unclear whether the postoperative analgesic effect of intraoperative DEX for patients with gastric ESD was dose-dependent. Second, the sample size of this study was evaluated based on the primary outcome, VAS pain score at 2 h postoperatively; because the sample size was small, this study may not be able to determine the between-group differences of some secondary outcomes, such as dosage of morphine in the ward, incidences of hypertension and bradycardia, and incidence of postoperative nausea and vomiting score ≥ 2 in the PACU. Third, this study only included healthy adults aged 18–65 years with ASA physical status grade I-II. The available evidence indicates that advanced age and low baseline arterial blood pressure are independent risk factors of hemodynamic instability in noncardiac ICU patients receiving intravenous DEX for sedation [42]. Thus, our results may not be extrapolated to patients aged > 65 years and to those with poor health status. Fourth, this study only included patients with gastric ESD, not those with esophageal and intestinal ESD procedures. The available literature shows that the incidence, severity, and features of postoperative pain are different among esophageal, gastric, and intestinal ESD procedures [8,43,44,45]. Thus, the findings of this study are also not suitable for patients undergoing non-gastric ESD procedures. Considering the fact that intravenous DEX can effectively decrease postoperative pain intensity, extend the postoperative pain-free period and reduce consumption of opioids for postoperative pain control in surgical patients with general anesthesia, we argue that more clinical studies to address the above issues are still required.

## 6. Conclusions

This study demonstrates that intraoperative DEX can effectively alleviate postoperative pain with a slightly reduced dosage of morphine, and decreased severity of postoperative nausea and vomiting after gastric ESD.

## Figures and Tables

**Figure 1 jcm-12-01816-f001:**
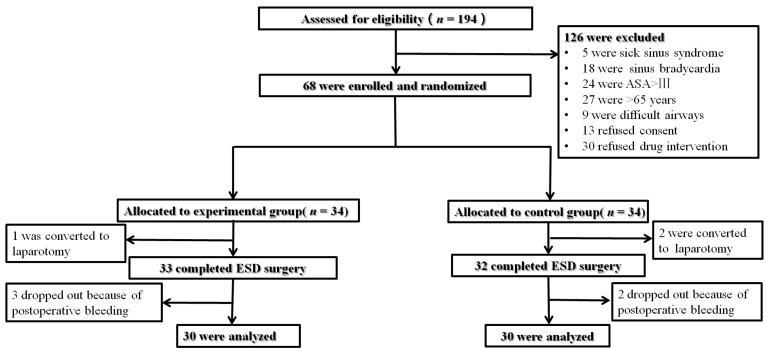
The flow chart of included and excluded patients. ASA: American Society of Anesthesiologists; ESD: endoscopic submucosal dissection.

**Figure 2 jcm-12-01816-f002:**
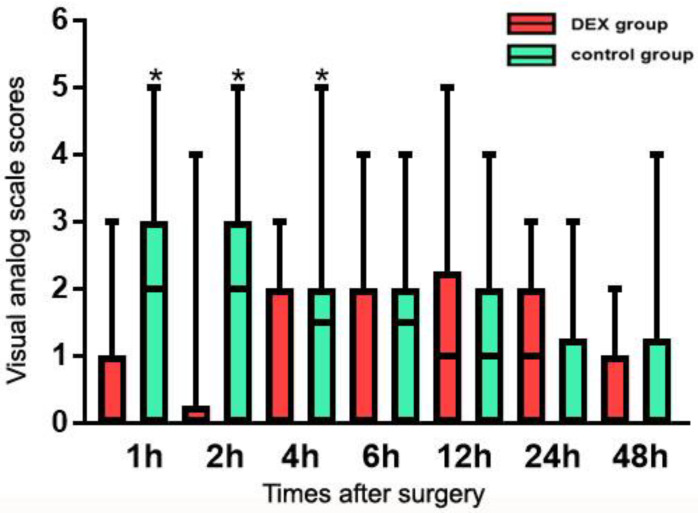
Postoperative pain scores. Notes: DEX, dexmedetomidine; Values are present as mean ± SD. * *p* < 0.05, compared with Dex group.

**Figure 3 jcm-12-01816-f003:**
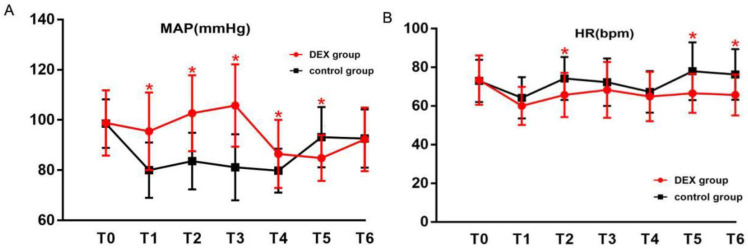
MAP (**A**) and HR (**B**) changes during the observable period. Values are present as mean ± SD. MAP, mean artery pressure; HR, heart rate; T0, before induction; T1, 1 min after induction; T2, at intubation; T3, at 5 min after intubation; T4, at the end of the procedure; T5, extubation; T6, 5 min after extubation. * *p* < 0.05, intergroup comparisons.

**Table 1 jcm-12-01816-t001:** Demographic data and clinicopathological characteristics of included patients.

Variables	DEX Group (*n* = 30)	Control Group (*n* = 30)	*p* Values
Age (years)	57.0 ± 7.0	55.8 ± 7.5	0.525
Sex (Male/female)	22/8	18/12	0.273
BMI (kg/cm^2^)	25.0 ± 3.5	24.3 ± 2.0	0.338
ASA (I/II)	8/22	6/24	0.542
Smoking	6 (20.0%)	8 (26.7%)	0.542
Alcohol use	8 (26.7%)	8 (26.7%)	1
Comorbidities			
Hypertension	10 (33.3%)	11 (36.7%)	0.787
Diabetes	2 (6.7%)	8 (26.7%)	0.080
Coronary heart disease	2 (6.77%)	2 (6.7%)	1
Hyperlipidemia	17 (56.7%)	17 (56.7%)	1
Repeated ESD procedure history	6 (20.0%)	8 (26.7%)	0.542
Specimen size (cm)	3.4 ± 1.2	2.8 ± 1.3	0.081
Tumor invasion depth			
Mucosa	22 (73.3%)	21 (70.0%)	0.774
Submucosa	8 (26.7%)	9 (30.0%)	
Localized sites			
Upper third	4 (13.3%)	8 (26.7%)	0.223
Middle third	11 (36.7%)	13 (43.3%)	
Lower third	15 (50.0%)	9 (30.0%)	
Histopathology			
Carcinoma	18 (60.0%)	18 (60.0%)	0.687
Dysplasia	3 (10.0%)	5 (16.7%)	
Others (leiomyomata/heterotopic pancreas)	9 (30.0%)	7 (23.3%)	

Values are present as number of patients (%), median (inter quartile range), or mean ± SD. Notes: Dex, dexmedetomidine; BMI, body mass index; ASA, American Society of Anesthesiologists; ESD, endoscopic submucosal dissection.

**Table 2 jcm-12-01816-t002:** Incidence of postoperative moderate to severe pain and dosage of morphine.

Variables	DEX Group (*n* = 30)	Control Group (*n* = 30)	*p* Values
Moderate to severe pain	8 (26.7%)	16 (53.3%)	0.035
Dosage of morphine in the PACU (mg)	0 (0, 0)	0 (0, 1.25)	0.000
Dosage of morphine in the ward (mg)	0 (0, 1)	0 (0, 1)	0.811
Total dosage of morphine within 24 h (mg)	0 (0, 1)	1 (0, 2.25)	0.009

Values are present as number of patients (%) or median (inter quartile range). Notes: DEX, dexmedetomidine; PACU, postanesthesia care unit.

**Table 3 jcm-12-01816-t003:** Anesthesia and intraoperative data.

Variables	DEX Group (*n* = 30)	Control Group (*n* = 30)	*p* Values
Hypertension	3 (10.0%)	1 (3.3%)	0.612
Hypotension	5 (16.7%)	16 (53.3%)	0.006
Tachycardia	3 (10.0%)	3 (10.0%)	1
Bradycardia	8 (26.7%)	4 (13.3%)	0.333
Ephedrine (mg)	0 (0, 0)	3.0 (0, 7.5)	0.001
Dosage of propofol (mg)	435 (283.8, 682.5)	497 (380, 622.5)	0.333
Dosage of remifentanil (µg)	852.5 (549.8, 1325.8)	906.5 (722.8, 1095.0)	0.579
Bleeding (mL)	10 (5, 15)	5 (5, 10)	0.509
Time to awakening (min)	7 (5, 10)	8 (5.75, 11)	0.484
Time to extubation (min)	10 (8, 14)	10 (8, 13)	0.976
Intraoperative fluids (mL)	300 (207.5, 355)	300 (207.5, 400)	0.323
Duration of anesthesia (min)	108 (72, 142.25)	93.5 (75.8, 135.5)	0.492
Duration of procedure (min)	62.5 (37.5, 101)	57 (33.5, 86)	0.437

Values are present as number of patients (%) and median (inter quartile range). Notes: DEX, dexmedetomidine.

**Table 4 jcm-12-01816-t004:** Postoperative data.

Variables	DEX Group (*n* = 30)	Control Group (*n* = 30)	*p* Values
Adverse events in the PACU			
Nausea and vomiting score ≥2	0 (0)	4 (13%)	0.112
Nausea and vomiting score			
Arrival in PACU	0 (0, 0.25)	1 (1, 1)	0.000
At 15 min	0 (0, 1)	1 (1, 1)	0.000
PACU discharge	0 (0, 1)	1 (1, 1)	0.000
Hypertension	0 (0.0%)	1 (3.3%)	1
Hypotension	6 (20.0%)	0 (0.0%)	0.024
Bradycardia	5 (16.7%)	1 (3.3%)	0.195
Dosage of ephedrine (mg)	0 (0, 0)	0 (0, 0)	0.001
Dosage of ondansetron (mg)	0 (0, 0)	0 (0, 0)	0.040
Length of PACU stay (min)	33.3 ± 5.0	35.7 ± 8.0	0.167
Patient satisfaction	10 (10, 10)	10 (9, 10)	0.210
Length of hospital stay (days)	10 (8, 12)	9 (7, 12)	0.567

Values are present as number of patients (%), median (inter quartile range), or mean ± SD. Notes: DEX, dexmedetomidine; PACU, Postanesthesia care unit.

## Data Availability

The data presented in this study are available on request from the corresponding authors.

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
