# Peer review of "Intraoperative Dexmedetomidine Decreases Postoperative Pain after Gastric Endoscopic Submucosal Dissection: A Prospective Randomized Controlled Trial"

_jcm, 2023, doi:10.3390/jcm12051816_

Round 1

Reviewer 1 Report

The article by Luo et al. reports on the efficacy of dexmedetomidine for postoperative pain after gastric ESD. This prospective, randomised, controlled trial was designed to evaluate the effect of intraoperative dexmedetomidine (DEX) on postoperative pain after gastric ESD. There have been studies on sedation and anaesthesia with dexmedetomidine for gastric ESD, but few have focused on postoperative pain. However, postoperative pain in gastric ESD is influenced by several factors that need to be fully investigated.

Comments on the study are as follows.

Main:

1. Postoperative pain after gastric ESD is often attributed to the operation time, the use of CO2 insufflation and the burning effect on the muscle layer. Although the background of the target tumours is discussed in this study, the surgical details that influence postoperative pain have not been adequately investigated.

2. In this study, the VAS score for postoperative pain at 2 hours was used as the primary outcome; in ESD with CO2 insufflation, abdominal distention and mild pain are common symptoms within 2 hours after surgery, but most resolve spontaneously within half a day. The proportion of ESD cases treated with CO2 insufflation should be discussed in detail in this trial.

3. Does the VAS score for pain include post-extubation pain, IV pain, etc?

Author Response

Reviewers 1

  1. Postoperative pain after gastric ESD is often attributed to the operation time, the use of CO2 insufflation and the burning effect on the muscle layer. Although the background of the target tumours is discussed in this study, the surgical details that influence postoperative pain have not been adequately investigated.

Responses: Thank you for your good questions. In this revision, we have added the details of gastric ESD procedures as follows. In this study, gastric ESD was performed by standard procedures including making, submucosal injection, incision of lateral margin of tumor, dissection of the tumor and electrocoagulation hemostasis. The marks were firstly made 5 mm surrounding the lesion mucosa using a needle knife. Then, the mixture of epinephrine (1:100,000 solution in saline) and indigo carmine was injected into the submucosal layer around the marks. After the lesion mucosa was removed with an electrosurgical knife from the marks, hemostasis was performed with hemostatic forceps or a clutch cutter. For each patient, carbon dioxide insufflation was used during gastric ESD and stopped immediately at the end of procedure. Seeing lines 135-144”on Page 5.

Considering that experience of the endoscopists is a main factor that influences surgical procedures and postoperative pain, all gastric ESD procedures were performed by the endoscopists with over 5-year experience in ESD and more than 500 ESD surgeries. Furthermore, surgical dissection depth is also a contributing factor of postoperative pain. As surgical dissection depth depends on tumor invasion depths, thus, we classed the tumors into mucosa and submucosal types according to their invasion depths. In Table 1 of our paper, we had provided these possible factors that affected occurrence of postoperative pain, such as gender, duration of anesthesia, operative time, location, depth, and pathological classifications of the lesions.

  1. In this study, the VAS score for postoperative pain at 2 hours was used as the primary outcome; in ESD with CO2 insufflation, abdominal distention and mild pain are common symptoms within 2 hours after surgery, but most resolve spontaneously within half a day. The proportion of ESD cases treated with CO2 insufflation should be discussed in detail in this trial.

Responses: Thank you for your good suggestion. We completely with you that mild pain are common symptoms within 2 hours after ESD surgery and resolved within half a day. However, moderate to severe pain is still present in some patients after gastric ESD and cannot alleviate without appropriate management. Indeed, CO2 insufflation is one of important factors that cause postoperative pain. According to our routine practices, CO2 insufflation was used for each patients undergoing gastric ESD in our study. These contents have been added in method section of revised paper. Seeing lines 143-144”on Page 5.

  1. Does the VAS score for pain include post-extubation pain, IV pain, etc?

Responses: Thank you for your question. In our study, we only assessed the upper abdominal pain using VAS score, did not evaluate post-extubation pain and IV pain.

Reviewer 2 Report

I have several comments on the manuscript. Were the patients really intubated in the lateral position? Was there a difference between moderate and severe pain between the groups? In the results, both pain intensities are described together. From the graph it seems that the pain intensity was up to VAS 5, i.e. only moderate. From table 2 it seems that almost no patients in the PACU received morphine, or only 1 mg; can we really conclude that p=0.000? The same applies to the morphine dose of the ward. Only the morphine dose comparison, or number of morphine doses, should be reported in the results. Comparing the number of patients is misleading. What were the criteria for morphine dose administration? If all patients should have the same criteria, why is it not only the dose that differs but also the VAS? Logically, all patients should have the same pain intensity with the same treatment approach. How was ephedrine administered? The table shows a total dose of 3 mg in the DEX group, yet it was given in boluses of 6 mg. The conclusions should be critically evaluated. The study showed a slight reduction in pain or morphine dose in postoperative analgesia, but also significant side effects on blood pressure. The reduction in morphine doses was statistically rather than clinically significant. 

Author Response

Reviewers 2

  1. Were the patients really intubated in the lateral position?

Responses: Thank you for your question. Yes, all patients were really intubated in the lateral position. As you know, gastric ESD is routinely performed in the lateral position. In order to ensure that the patient is comfortable during the operation, we usually put the patient in a lateral position that they consider comfortable in awake status before anesthesia. As all intubation were performed with attending anesthesiologists with abundant experience of airway management in lateral position, no failed intubation and adverse events associated with airway management occurred in any patient. These contents have been added in method section of revised paper. Seeing lines 124-126”on Page 5

  1. Was there a difference between moderate and severe pain between the groups? In the results, both pain intensities are described together. From the graph it seems that the pain intensity was up to VAS 5, i.e. only moderate. From table 2 it seems that almost no patients in the PACU received morphine, or only 1 mg; can we really conclude that p=0.000? The same applies to the morphine dose of the ward. Only the morphine dose comparison, or number of morphine doses, should be reported in the results. Comparing the number of patients is misleading.

Responses: Thank you for your questions. In the results, we had provided the details regarding the occurrence of postoperative pain, such as “The incidence of moderate to severe pain was 26.0% and 53.3% in the DEX and control groups, respectively, with a significant between-group difference (P<0.05). Seeing line 235-237 in Page 9”. In our study, for data with a non-normal distribution, they were presented as medians (inter quartile range, IQR). IQR indicates the 25 to 75% quartiles of the data not including minimal and maximal values. The intergroup comparisons of data with a non-normal distribution were performed by the Wilcoxon signed-rank test. Our results showed that P value of between-group comparison for dosages of morphine in PACU was 0.000. According to your advices, only the morphine dose comparison was reported in the results of this revised paper. The contents regarding the results of between-group comparison in the number of patients needing morphine in the text and table 2 have been deleted.

  1. What were the criteria for morphine dose administration? If all patients should have the same criteria, why is it not only the dose that differs but also the VAS? Logically, all patients should have the same pain intensity with the same treatment approach.

Responses: Thank you for your questions. In method section, we had described the criteria for morphine administration, i.e., if the VAS pain score was ≥ 4 or patient needed additional analgesia, 1 mg intravenous morphine was administered. Seeing line 159-160 on page 6.

Indeed, all patients included in this used the same criteria of morphine administration for postoperative pain control. However, it must be noted that sensitivity to pain is significantly different among patients. That is, patients with the same pain intensity may need different dose of morphine for postoperative pain control and the same dose drug may produce different analgesic effect in various patients. This is why individualized treatment is often emphasized to achieve adequate postoperative pain control in clinical practice.

  1. How was ephedrine administered? The table shows a total dose of 3 mg in the DEX group, yet it was given in boluses of 6 mg. The conclusions should be critically evaluated. The study showed a slight reduction in pain or morphine dose in postoperative analgesia, but also significant side effects on blood pressure. The reduction in morphine doses was statistically rather than clinically significant. 

Responses: Thank you for your questions. Regarding ephedrine administration, in method section, we described as follows: If hypotension persisted for more than 2 min and was no respond to a load therapy of 200 ml fluid, intravenous ephedrine 6 mg was given. Seeing lines 187-189 on page 7.

In table 3, 3 mg of ephedrine in control group was median of ephedrine dosage, rather a bolus dose. Because the 15th is 0 mg and the 16th is 6 mg, the calculated median was 3 mg.

Our study was the first to investigate the effect of dexmedetomidine on postoperative pain in patients undergoing gastric ESD and showed that dexmedetomidine resulted in a 50% decrease in the incidence of moderate to severe postoperative pain. The reduction in morphine doses was very small, but it may be clinically significant for patients undergoing ESD and needing early discharge as the lower morphine consumption can produce a lower level of sedation and a decreased risk of adverse events. This is agreement with the basic principles of enhanced recovery after surgery protocols for gastric ESD (Li J, Kang G, Liu T, Liu Z, Guo T. Feasibility of Enhanced Recovery After Surgery Protocols Implemented Perioperatively in Endoscopic Submucosal Dissection for Early Gastric Cancer: A Single-Center Retrospective Study. J Laparoendosc Adv Surg Tech A. 2023; 33(1):74-80.). according to your suggestions, we have revised the conclusions as follows: Intraoperative DEX can significantly decrease postoperative pain level, with a slightly reduced dosage of morphine and decreased severity of postoperative nausea and vomiting after gastric ESD.

Indeed incidence of postoperative hypotension was significantly higher in DEX group. However, dexmedetomidine administration provided a decreased risk of intraoperative hypotension. These issues have been clearly described in the results and explanted in the discussion section. Seeing lines 314-335 on pages 11-12.

Reviewer 3 Report

Greetings

I read your study. The study methodology is good, and the manuscript is written and presented well. The results are very much expected. 

While I do not have major flaws and critics, I can find that basal postoperative analgesia is not mentioned. Were any NSAIDs or paracetamol, or any block considered as per multi-modal (recommended) techniques? 

If not, it is a limitation of the study. 

Best of luck

Author Response

Reviewers 3

  1. While I do not have major flaws and critics, I can find that basal postoperative analgesia is not mentioned. Were any NSAIDs or paracetamol, or any block considered as per multi-modal (recommended) techniques? 

Responses: Thank you for your questions. In the method section, we had described basal postoperative analgesic and antiemetic scheme of patients as follows: At the end of procedure, tramadol 50 mg and ondansetron 4 mg were intravenously given as analgesic and anti-emetic drugs, respectively. These are our routine managements after ESD surgery. Seeing line 147 on page 6.

Reviewer 4 Report

This is a comparison of dexmedetomidine with control group for post operative pain after ESD procedures. It is a well established fact that intraoperative decmedetomidine reduces postoperative analgesic requirement. As such the concept is not new and does not add to current knowledge. The sample size calculation requires more elucidation. VAS was observed as a whole number and becomes an ordinal data. Hence mean us not an appropriate statistical test. Median and rank test should be used for comparison. Introduction can be shortened and less references be used.

Author Response

Reviewers 4

  1. This is a comparison of dexmedetomidine with control group for post operative pain after ESD procedures. It is a well established fact that intraoperative decmedetomidine reduces postoperative analgesic requirement. As such the concept is not new and does not add to current knowledge.

Responses: Thank you for your questions. We completely agree with you that intraoperative dexmedetomidine reduces postoperative analgesic requirement. However, this conclusion is from studies of different surgical operations, rather ESD, a new kind of minimally invasive endoscopic surgery. As we have described in introduction, there is not study to determine whether intraoperative administration of DEX during ESD procedure with total intravenous anesthesia can attenuate postoperative pain level. This is main object of this RCT.

  1. The sample size calculation requires more elucidation. VAS was observed as a whole number and becomes an ordinal data. Hence mean us not an appropriate statistical test. Median and rank test should be used for comparison. Introduction can be shortened and less references be used.

Responses: Thank you for your questions. You are exactly right that VAS can be observed as a whole number and becomes an ordinal data. However, when VAS data are a normal distribution, they can also be used as measurement data for sample size calculation and statistical analysis, i.e., previous randomised controlled trial assessing the efficacy of various methods on postoperative pain after endoscopic submucosal dissection (Choi HS, et al. The efficacy of transdermal fentanyl for pain relief after endoscopic submucosal dissection: a prospective, randomised controlled trial. Dig Liver Dis. 2012; 44(11):925-9.). Similarly, in recent randomised controlled trial assessing the efficacy of various interventions on postoperative pain after surgical operation, such methods of sample size calculation and statistical analysis have been used. (Xu M, et al. Combined Ultrasound-Guided Thoracic Paravertebral Nerve Block with Subcostal Transversus Abdominis Plane Block for Analgesia After Total Minimally Invasive Mckeown Esophagectomy: A Randomized, Controlled, and Prospective Study. Pain Ther. 2023 Jan 17. doi: 10.1007/s40122-023-00474-5). If you consider that VAS must be present as an ordinal data, we will revise rewritten the contents of sample size calculation.

According to your suggestions, we have shortened introduction and reduced the related references. The word number of introduction section has been decreased from 514 in original version to 334 in this new version. Please you recheck the introduction. Because of introduction revision requirements, 2 references in the introduction of original version have been deleted. Thus, there are a total of 45 references in this new version.

Round 2

Reviewer 1 Report

The authors responded appropriately to the reviewers' comments. We therefore recommend acceptance of the paper in its current form.